# Extracellular Vesicles as New Players in Drug Delivery: A Focus on Red Blood Cells-Derived EVs

**DOI:** 10.3390/pharmaceutics15020365

**Published:** 2023-01-21

**Authors:** Sara Biagiotti, Faiza Abbas, Mariele Montanari, Chiara Barattini, Luigia Rossi, Mauro Magnani, Stefano Papa, Barbara Canonico

**Affiliations:** 1Department of Biomolecular Sciences, University of Urbino Carlo Bo, 61029 Urbino, PU, Italy; 2AcZon s.r.l., 40050 Monte San Pietro, BO, Italy

**Keywords:** extracellular vesicles, red blood cell extracellular vesicles, nanomedical drug delivery, drug delivery systems, cargo loading, therapeutic applications

## Abstract

The article is divided into several sections, focusing on extracellular vesicles’ (EVs) nature, features, commonly employed methodologies and strategies for their isolation/preparation, and their characterization/visualization. This work aims to give an overview of advances in EVs’ extensive nanomedical-drug delivery applications. Furthermore, considerations for EVs translation to clinical application are summarized here, before focusing the review on a special kind of extracellular vesicles, the ones derived from red blood cells (RBCEVs). Generally, employing EVs as drug carriers means managing entities with advantageous properties over synthetic vehicles or nanoparticles. Besides the fact that certain EVs also reveal intrinsic therapeutic characteristics, in regenerative medicine, EVs nanosize, lipidomic and proteomic profiles enable them to pass biologic barriers and display cell/tissue tropisms; indeed, EVs engineering can further optimize their organ targeting. In the second part of the review, we focus our attention on RBCEVs. First, we describe the biogenesis and composition of those naturally produced by red blood cells (RBCs) under physiological and pathological conditions. Afterwards, we discuss the current procedures to isolate and/or produce RBCEVs in the lab and to load a specific cargo for therapeutic exploitation. Finally, we disclose the most recent applications of RBCEVs at the in vitro and preclinical research level and their potential industrial exploitation. In conclusion, RBCEVs can be, in the near future, a very promising and versatile platform for several clinical applications and pharmaceutical exploitations.

## 1. Extracellular Vesicles: An Overview of Their Origin and Composition

Although the following paragraphs will provide a general introduction to extracellular vesicles (EVs), hints on their isolation, their characterization, and their journey to the different body districts, the main purpose of the present review is to describe EVs as efficient drug carriers. In fact, pharmacologic molecules are protected by EV membranes from proteases, nucleases, pH and osmolality fluctuations, and other external factors. Specifically, we will frame red blood cells as sources of EVs (RBCEVs) that can address the major requirements for efficient drug-delivery, providing a useful and insightful description of procedures (also patented) to produce RBCEVs, with their advantages and limitations.

EVs are cell-derived membrane vesicles that represent an endogenous mechanism for intercellular communication [1]. The original classification distinguished exosomes (nano-sized vesicles with a diameter in the range of 30 to 120 nm) [2], originating from the formation of multivesicular bodies, microvesicles, which are formed by cell membrane budding, and apoptotic bodies, derived from dying cells. This classification has been recently revised by the International Society of Extracellular Vesicles (ISEV), establishing the need to use the term EVs for all the subtypes of vesicles, given that they overlap in size and this can cause confusion [3]. Therefore, the ISEV entitled EVs smaller than 200 nm as “small EVs” and those larger than 200 nm as “large EVs” [3]. Consequently, in depth characterization has led to better categorize EVs on the basis of size, density, biochemical composition, cellular origin (e.g., oncosomes derived from tumour cells) or treatment condition (e.g., hypoxic EVs) [4,5]. As cited, small EVs (exosomes) range between 50 and 150 nm, whereas large EVs (microvesicles) and apoptotic bodies range between <1 µM and 1–5 µm in diameter, respectively [4].

### 1.1. EV Biogenesis

The differences in size could represent an indication of the specific biogenesis processes. Biogenesis and secretion of EVs are controlled by specific proteins such as GTPases and lipids, as genetic studies have proven. The Rab family of small GTPases plays an important role in intracellular trafficking, and several Rabs play a significant role in EVs release, particularly Rab27a, Rab27b, Rab35, and Rab11 [6,7,8,9,10,11,12]. EVs production and release can be modified or orchestrated by internal cellular processes or external stimuli. For example, the modulation of intracellular calcium levels in cancer cells [13] is a possible factor increasing EVs production. Cargo selection, packaging, and compartmentalization are processes regulated at multiple levels. Briefly, the endosomal sorting complex required for transport (ESCRT)-dependent pathway is involved in the selection and distribution of proteins within exosomes [14]. CD63 is a molecule involved in sorting EVs cargo [15], and it guides cargo selection in MVs [16]. However, despite all the knowledge acquired so far, the biogenesis and sorting mechanisms of extracellular vesicles are still unclear. However, exosomes are well-characterized EVs normally generated by the internal budding of endosomes, thus producing multivesicular bodies (MVBs) and ultimately forming intraluminal vesicles (Figure 1). Such vesicles fuse with the cell membrane, releasing the so-called exosomes into the extracellular environment [17]. Besides the production of MVBs, the biogenesis pathway of EVs can also be associated with secretory autophagy [18,19,20,21]. Although the current orientation is dictated by the ISEV, which we shared in the initial part of our review, we must continue to differentiate EVs based on their different origins.

The biogenesis of MVs is entirely derived from the plasma membrane and shares many of the same proteins involved in exosome biogenesis [22,23,24]. This kind of smaller EVs (exosomes), as well as MVs, can transfer bioactive molecules, including proteins, DNA, mRNA, and miRNA, and these cargoes are able to modify the extracellular milieu and recipient cells, both proximally and distally [25,26]. MVs are generated from sites of high membrane blebbing; indeed, their biogenesis involves the vertical trafficking of loaded molecules to the plasma membrane along with the reorganization of membrane lipids. Furthermore, contractile machinery at the surface allows vesicle budding [27] (Figure 1). The ESCRT complex also plays an important role in MV biogenesis. Moreover, ARF-6 is involved in the trafficking of cargo to the cell surface in MVs [28]. Several proteins such as TSG101, ALIX, and ARRDC1 assisted in MV release, and ESCRT-III and ALIX are involved in cytokinetic abscission [6,29]. A significant role in MV structure is played by the extracellular concentration of calcium; in fact, it was reported [30] that a high level of calcium not only induces membrane phospholipid disarrangement but also increases the level of vesiculation to produce more MVs [31], especially in erythrocytes and platelets [32]. Finally, curving of the plasma membrane at the cell periphery and lateral pressure created by protein–protein interactions also play crucial roles in de novo MV formation [33].

### 1.2. EVs Benefits: Their Journey to the Different Body Districts

The combination of exosomes and nanotechnology can facilitate the development of next-generation theragnostic nanoplatforms. Extracellular vesicles are highly biocompatible and relatively less toxic due to their natural origin. Indeed, EVs can escape immune clearance, which makes them highly stable in blood circulation. These properties make them preferable drug delivery carriers compared to synthetic vehicles or nanoparticles [34,35,36,37,38]. Moreover, there is a clear indication that EVs can cross multiple biological barriers, as demonstrated by several studies of neuronal EVs in the cerebrospinal fluid, blood, urine, and tears [39]. However, delivery to the specific target mainly depends on the surface marker on specific cells [40,41]. These characteristics amplify their target potential; in fact, among the physiological barriers EVs cross, the blood–brain barrier (BBB) opens the possibility that they may be enriched in a specific neuron population. Moreover, naturally occurring molecules in them can be encapsulated with drugs, thus producing a synergic effect [42,43]. Being a highly stable structure, EVs can circulate systemically and have been identified in blood, urine, cerebrospinal fluid, saliva, milk, and tears [39,44,45,46,47,48,49,50,51]. EVs possess an excellent safety profile in both animal models and clinical trials. Different studies reported data on the immunogenicity of EVs from different sources. For example, administration of EVs derived from human embryonic kidney (HEK) 293 T cells exhibited very little toxicity and immune response in healthy mice [52,53]. EVs cargo remains stable even in the highly acidic environment of the stomach, as illustrated by the reduction in delayed-type hypersensitivity by orally supplemented miRNA-150 EVs [54]. Proteomics analysis of the urinary EVs revealed the presence of specific EVs from glomerular endothelial cells and mesangial cells in healthy individuals [55], which truly indicates that endogenous EVs have glomerular filtration barrier (GFB)-penetrating capacity (Figure 2). More tracing evaluation is still needed to clarify the distribution of exogenous EVs; furthermore, the internalization process of EVs is closely dependent on their origins and the type of recipient cells. Exogenously administered EVs possess and display specific biodistribution profiles, natural cell-targeting abilities, and pharmacokinetics due to their peculiar components. Indeed, certain EVs also reveal intrinsic therapeutic characteristics, in the context of regenerative medicine [56]. Furthermore, some EVs display characteristics of tropism for a particular cell or tissue. This feature could be exploited to deliver drugs to specific targets while avoiding off-target effects [57]. The natural targeting properties of EVs can be modulated by their compositions in lipids, integrins, and tetraspanins. Engineering EVs leads to altering these targeting properties in several ways: by the addition of targeting moieties anchored via the phosphatidylserine-binding C1C2 domains of lactadherin [58], the expression of lysosome-associated membrane protein 2 fusion proteins [40], glycosylphosphatidylinositol-anchored targeting moieties [59], and transferrin-conjugated magnetic particles bound to the transferrin receptor expressed on EVs [60] must reach their activity sites within the cell in adequate amounts [61].

In the next topic, specific details will be added to better understand the mechanisms that mediate the EVs’ application as tools for mediated drug delivery.

### 1.3. Extracellular Vesicles (EVs): A Novel Drug Delivery System

We have profusely described EVs as entities carrying diverse cargo, including lipids, proteins, long non-coding RNA (lncRNA), and microRNA (miRNA), that can be transferred to recipient cells to mediate intercellular communication. These naturally occurring nanovesicles are released by different types of cells, including reticulocytes, mesenchymal stem cells, T cells, B lymphocytes, NK cells, dendritic cells, and some tumors. There is strong evidence that extracellular vesicles are involved in both pathological and physiological processes, including cellular homeostasis, infection propagation, cancer development, and cardiovascular diseases [62,63,64,65].

Due to their innate function in cell-to-cell communication, EVs can be used effectively for drug delivery. The biggest advantage of EVs as drug delivery vehicles is probably that EVs can be taken from an organism and returned to the same organism in vivo after being loaded with therapeutic agents, which are thought to be non-immunogenic [66]. Another advantage is that EVs can carry molecules through physiological barriers, such as the blood–brain barrier, which are hard to cross using conventional delivery methods; this is particularly true when using nucleic acids as cargo [67,68,69]. When exogenous RNAs are directly introduced into the body, they are normally degraded by nucleases or filtered in the kidneys before reaching the target site. Various studies have suggested that both coding and noncoding RNAs can be transported through EVs. In addition, microRNAs can be transferred to different cell types via EVs. Thus, EVs can be considered an efficient RNA-based drug delivery carrier. Several therapeutic applications include gene therapy, targeted therapy, vaccination, and the treatment of kidney and autoimmune diseases [70].

### 1.4. Other Actors in Next Generation Drug Delivery Platforms: Taking a Glance

Although our review is centered on EVs, and in particular, EVs derived from RBCs, we just want to mention other nanosized drug delivery systems (DDS). Liposomes were discovered nearly 55 years ago, and they currently represent a staple in the field of drug delivery. Among nanoDDS, liposomes are small spherical structures [4] that could be modified with different active targeting agents [71]. They displayed proper bioavailability, long half-lives, size-control measures, low risk-to-benefit ratios, and control release features [72]. Our experience [73] with these entities pointed out several advantages of intracellular internalization in triple-negative breast cancer cells (MDA-MB-231) for sucrose-decorated liposomes loaded with berberine hydrochloride. Recently, liposomes demonstrated their significant efficacy in SARS-CoV-2 vaccines [74]. Such formulations are paving the way for future success in several therapeutic applications [74]. The influence of lipid composition and particle size on pharmacokinetics is common to both liposomes and EVs; in addition, EVs are advanced drug delivery entities containing several proteins that may contribute to their pharmacokinetic behavior in vivo [75].

The development of mesoporous materials has prospected their use as drug delivery systems. The well-known textural properties of these materials have inspired their translation to nanoscale constructs, resulting in mesoporous silica nanoparticles (MSNPs) [76]. Our experience is specifically related to fluorescent silica nanoparticles (SiNPs), which appear to be a promising imaging platform and show a specific subcellular localization that is mainly at the mitochondrial level. We conjugated SiNPs to one of the most commonly used anticancer drugs, doxorubicin, and we tested these functionalized SiNPs (DOX-NPs) on the breast cancer cell line MCF-7 [77]. Generally, carrier nanoparticles (NPs) reveal a poor penetration capacity and an inadequate balance between drug retention in the bloodstream and drug release at the specific pathologic body districts [78]. These findings represent an obstacle in drug release that EVs could overcome.

Recently, nanocellulose has attracted considerable attention for its applications in drug delivery platforms; this is mainly due to its biocompatibility, large specific surface area, good mechanical strength, stiffness, and renewability [79]. These characteristics qualify nanocellulose as a material with good drug loading and binding capacities. Khalid and co-workers reported that different types of nanocellulose-based materials such as single, hybrid, or nanocomposite systems have been fabricated for application in drug delivery systems [80]. Nonetheless, it is difficult for the human body to degrade nanocellulose-based materials, and the interaction mechanism between nanocellulose and cells is still unclear. These issues are obviously lacking in EVs, due to their biologic features.

Finally, in the field of cancer immunotherapy and tumor microenvironment remodeling, two different nanostructures will be cited. The first, formulated by Liang et al., represents a novel nanovaccine based on antigen self-presentation and immunosuppression reversal, named ASPIRE. The ASPIRE formulation has a nanoscale size and good stability; its preparation arises from dendritic cells (DCs). Briefly, the nanovaccine is derived from recombinant adenovirus-infected DCs, and anti-PD1 antibodies, MHC-I molecules, and B7 co-stimulatory molecules are concomitantly attached to the membrane of mature DCs through cell reprogramming. Then, Dcnv-rad-ag was separated by multi-step density gradient supercentrifugation. The novel nanovaccine agent ASPIRE greatly improves the efficiency of antigen presentation, activating both naive T cells and exhausted T cells, and improving anti-tumor immunity, finally representing an important progress in tumor immunotherapy [81].

The second nanosized construct, by Cheng and co-workers, is essentially represented by glucose-contained radical micelles for synergistic photoimmunotherapy and aims for dual tumor microenvironment remodeling. They designed a novel amphiphilic copolymer, glucose-PEO-b-PLLA-TEMPO, to encapsulate clinical therapeutics (CUDC101 and photosensitizer IR780), providing a promising strategy to integrate small molecule immune checkpoint inhibition and photodynamic therapy [82].

The above-mentioned issues can by no means exhaust such vast and complex topics, and they do not represent the focus of this review. Nevertheless, this short summary not only highlights other actors (besides EVs) in next-generation drug delivery platforms but also the involvement of our groups in the vast and vital area of drug delivery technologies.

### 1.5. Different Extracellular Vesicle Preparations/Isolations and Characterization: An Overview

Even if the International Society for Extracellular Vesicles (ISEV) began to publish the Minimal Information for Studies of Extracellular Vesicles (MISEV) guidelines in 2014, with regular updates, nowadays, no general consensus has been reached on the best method for EVs isolation and characterization, and the best option is often related to the starting volume of the EV source and to the processes to be applied downstream [5]. Different techniques have been applied to quantify and characterize EVs: Figure 3 highlights Western blot (WB), flow cytometry (FC), dynamic light scattering (DLS), electron microscopy (TEM and SEM), nanoparticle tracking analysis (NTA), and tunable resistive pulse sensing (TRPS). These methodologies allow analyzing EVs, focusing on different EV features such as: diameters and morphology (TEM), EV dimensions and concentrations (NTA), particle concentration (TRPS), size distribution and polydispersity (DLS), specific EV markers (WB), size, particle concentration, and specific EV marker (FC). Since there are many reviews that deal with methods of EV analysis, in this paragraph we will synthetically focus on the different methodologies for their isolation/obtaining, and subsequent characterization [83,84,85,86,87].

According to the literature, the most employed method to isolate EVs is differential ultracentrifugation, which has been considered the gold standard for a long time [88]. This method includes several centrifugation steps at increasingly longer times and higher speeds (up to 120,000× *g*). Based on Marassi et al.’s work, to pellet large extracellular vesicles, the biological fluid was centrifuged at 10,000× *g* for 30 min and 18,000× *g* for 30 min, then the resulting pellets from both centrifuges were united and resuspended in phosphate buffer followed by ultracentrifugation at 20,000× *g* for 30 min. To collect small extracellular vesicles, large extracellular vesicle-depleted supernatant was ultracentrifuged at 100,000× *g* for 70 min [89]. Among the advantages listed for using this technique are its high versatility due to the possibility of adjusting centrifuge parameters according to a scientist’s needs and good final yield. On the other hand, the high centrifuge speed might affect the intactness of the EVs and cause massive aggregation due to the presence of other proteins in the sample [90]. The application of an iodixanol/sucrose density gradient or a sucrose cushion to the high-speed ultracentrifugation significantly improved the quality of the isolated EVs [91]. The iodixanol/sucrose gradient exploits the similarity of exosome density (1.08 to 1.19 g/mL) to that of sucrose and iodixanol, which form a cushion preserving the integrity of EVs and separating high-density contaminant proteins (1.35 g/mL). In this method, the sample is poured onto the density gradient and then centrifuged at high speed. During the centrifugation, vesicles filter through the sucrose/iodixanol gradient until the point at which their density is equal to the gradient. With the employment of the sucrose cushion, vesicles are laid down on a high-density sucrose matrix with lower stress applied to EVs [92].

In recent times, size exclusion chromatography (SEC) has caught on in the multitude of EVs purification protocols. This technique is broadly used for the separation of macromolecules on the basis of their molecular size or hydrodynamic volume. Bigger species are eluted earlier than smaller species, which spend more time crossing the pores typical of the employed resins. Basically, the most widely used protocols include a preliminary ultracentrifugation purification step followed by sample loading on a qEV size exclusion column (Izon Science, New Zealand). To elute EVs from the column, phosphate buffer is used. After protein and particle quantitation, the richest fractions are concentrated using four 10 kDa nominal molecular weight centrifugal filter units [93,94]. Even this technique is not free from drawbacks in terms of purity: species featuring the same size (but different origin) are contemporaneously eluted; in addition, EVs are diluted during the procedure, requiring a supplementary concentration step according to the following activities [1]. Moreover, SEC requires considerable hands-on time for the preparation and maintenance of the isolation column [95,96,97].

Immuno-capture-based approaches represent an effective alternative to obtaining homogeneous EV samples by exploiting the interactions between proteins, naturally present in EVs, and their respective ligands. Duijvesz and co-workers developed a highly sensitive time-resolved fluorescence immunoassay (TR-FIA) for the capture/detection of EVs using anti-human CD9- or CD63-coated plates [98]. The high specificity of this approach is generally recognized, but, on the other hand, it is limited by the availability of reliable and cost-effective monoclonal antibodies, or aptamers, specific for EV isolation (often targeting tetraspanin proteins) and also by the necessity of a mild elution method that preserves EV integrity [99,100,101,102].

Ultrafiltration (UF) uses membranes with a molecular weight cut-off ranging from 10 to 100 kDa to isolate EVs from relatively diluted samples such as urine or blood. It is a quite rapid procedure lasting about 20 min (against several hours spent during ultracentrifugation), in which particles smaller than the membrane cut-off size pass through the membrane itself. This method is affected by a considerable quantity of protein contaminants and EV loss. Near conventional UF, tangential flow filtration (TFF) is utilized for EVs separation. In TFF, the sample moves tangentially across (not directly through) the hollow fiber membrane, which allows smaller molecules to pass while retaining bigger ones. One of the main advantages of using TFF instead of SEC is the higher final concentration of EVs [103,104].

Recently, asymmetrical flow field-flow fractionation (AF4) became one of the most used subcategories of flow field fractionation to separate EVs. In AF4, the separation force is generated by a cross-flow field inside the channel, which is divided from the main parabolic flow pumped through the channel and is directed through a semipermeable membrane that is located at the bottom (the accumulation wall). The membrane pores’ size prevents the EVs from passing through but allows smaller contaminants to exit the AF4 channel. The sample fractions are eluted out of the channel in the direction of the detectors by the remaining channel flow. Smaller EVs remain far from the accumulation wall due to their higher diffusion coefficient, in contrast to larger ones. One of the main advantages is due to the absence of stressful forces applied to the sample, on the other side, even in this case, the eluted sample results are diluted and need to be concentrated before being applied in further applications [105,106].

Polyethylene glycol (PEG) is one of the most used compounds present in commercial kits to cause EV precipitation. This isolation method assures a high recovery along with an important concentrative effect, but on the other side, it is highly recommended to isolate EVs before applying the precipitation protocol in order to avoid the massive presence of contaminants [107]. Habitually, an overnight incubation of 4 °C with 10% (*w*/*v*) PEG 8000 is used to precipitate EVs [108]. Gallert-Palau used the PRotein Organic Solvent PRecipitation (PROSPR) method to remove soluble proteins from plasma via precipitation in cold acetone with satisfactory results [109].

The analysis of potential showed that EV membrane components are negatively charged. Charge-based isolation techniques exploit this attitude to facilitate the interaction between EVs and positive entities. Ion-exchange chromatography, metal-affinity systems, or positively charged proteins, such as protamine, trap EVs, which can be released from the positively charged matrix by adding a high salt concentration to increase ionic strength [110,111].

In addition to the above-mentioned isolation methods, other techniques, especially those based on the lab-on-chip approach [112], were applied during the last few years, among which were acoustofluidic methods [113], capillary zone electrophoresis [114], conductive polypyrrole nanowire arrays [115], lateral displacement arrays [116], and viscoelasticity-based microfluidic systems [117].

Despite the increase in the development of advanced EV isolation techniques to obtain pure and high-quality EVs, there is currently no consensus on a suitable EV isolation method. This lack of agreement mainly depends on the damages caused by many of these protocols on isolated EVs, making downstream applications difficult. Since different techniques previously described exploit different principles to separate EVs from the rest of the sample macromolecules, the combination of complementary methods is an increasing trend in the literature [118,119,120].

During the last few decades, even the development of new methods for EVs characterization has increased a lot, but in the same manner as the isolation method, no general consensus has been reached, and the selection of the characterization procedure strongly depends on the EV source, isolation method, and downstream processes.

Electron microscopy can achieve a sufficient resolution (about 1 nm, since the electron wavelength is much shorter than the visible light wavelength) to allow the morphological study of extracellular vesicles, and, for this reason, it has been considered the gold standard for EV characterization for a long time. In transmission electron microscopy (TEM), the samples need to be dehydrated during their preparation to allow fixation with glutaraldehyde and placed in a vacuum environment to focus the electron beam directly on the sample. These requirements often cause artifacts in the analyses. As an alternative, cryo-EM is employed. It is a particular subtype of TEM where samples are rapidly frozen to preserve their morphology. To couple the morphological analysis in EM to EV immunophenotyping, it is possible to employ gold colloidal conjugated antibodies [121,122].

Atomic force microscopy (AFM) is a surface-based imaging technique allowing topographical imaging at sub-nanometer resolution (lateral resolution ≈ 3 nm and vertical resolution below 0.1 nm). It is based on the interaction between the probing sharp tip, mounted on a cantilever, and the sample surface. The most used technique for soft objects, such as EVs, characterization is the tapping mode in which the tip directly scans the surface of the sample. Furthermore, using specific antibody-coated surfaces, EV subpopulations can be identified. Unfortunately, even in this case, the preparation of the sample on a highly flat surface, such as mica, might modify the shape of EVs [123,124].

Dynamic light scattering (DLS) analyzes the fluctuation, due to Brownian motion, of the light scattered by nanomaterials and therefore the EVs, when illuminated with a laser beam in solution. The EV size, intended as hydrodynamic radius, and density are obtained using the Stokes–Einstein equation starting from their velocity distribution due to Brownian motion. The sensitivity of DLS ranges from 1 nm to 6 µm and does not require sample pre-treatment, but, in heterogeneous samples, the light scattered by larger particles covers the signal from smaller ones. For this reason, DLS employment in polydisperse samples is limited [125].

Nanoparticle tracking analysis (NTA) exploits the same DLS principle but uses a camera as a detector instead of a photon detector, allowing for the visualization of EVs ranging from 60 to 1000 nm. This feature promotes the use of NTA in place of DLS, especially in highly polydisperse samples. Likewise, DLS or even NTA is a technique requiring a relatively short time (less than one hour) and no sample preparation, but, exploiting the same principle as DLS, even in this case, a few large particles might obscure a multitude of smaller particles [126,127].

Flow cytometry (FC) is a powerful method to detect and characterize surface and/or cytoplasmic protein expression in EVs. In FC particles pass, individually, through a laser beam scattering light and emitting fluorescence signals to different channels. Conventional FC allows the detection of protein on relatively large entities (≥300 nm) and, often, EV scattering intensities are below the background noise. Besides, even the difference in antigen exposed on the surface between cells (>1000) and EVs (<100) needs to be considered before employing this characterization technique. To improve the limit of conventional FC detection, EVs are conjugated to micrometer-sized fluorescent latex beads carrying specific antibodies against membrane EV antigens. Recently, enhanced FC instruments have been developed to detect particles smaller than 300 nm; in particular, in nanoscale FC, thanks to the improvements in optical and fluidic systems, the threshold has been lowered to 100 nm [128,129].

Western blotting is frequently used to detect and quantify the presence of specific proteins in EVs that have been previously purified and disrupted to extract those proteins. In this method, EVs are lysed, and proteins from the lysate are run on an SDS-PAGE and, subsequently, blotted on a nitrocellulose membrane. The proteins in the membrane are then recognized using primary antibodies against the proteins themselves and secondary antibodies conjugated to revelation enzymes. As EVs are heterogeneous populations, there is no panel of proteins recognized as an exclusive EV marker. According to MISEV, at least one transmembrane protein and one cytosolic protein need to be detected to assess the EV presence in a complex mixture [130].

Near to Western blotting, even enzyme-linked immunosorbent assay (ELISA) is commonly used to detect and quantify EV proteins, in particular in its sandwich variant. In this assay, the plate is coated with capture monoclonal antibodies directed against an EV protein of interest. EV lysates are added to the plate, and the presence of the protein of interest is assessed by a second monoclonal antibody, against a different epitope, conjugated with a revealing enzyme. This assay requires low sample volume and more samples, than in Western blotting, can be contemporary analyzed due to the employment of, at least, 96-well plates. Unfortunately, intra- and inter-assay variability should be considered when employing this kind of test [100,131].

Resistive pulse sensing (RPS) measures the size and concentration of particles in a solution by measuring the change in conductance across a sensing pore upon passage of a particle, exploiting the Coulter principle. This system does not distinguish between EV and non-EV particles, and, in addition, the pore is prone to clogging by aggregates or sticky proteins naturally present in biological samples [132].

It is well known that the specific EV cargo is fundamental for cellular response upon EV delivery. In addition, understanding the specific molecular machinery that regulates the EV cargo intracellularly is necessary to understand the role of EVs in physiology and pathophysiology and their possible therapeutic use [133,134].

High-throughput technologies allow a multifaceted characterization of EVs content by identifying a variety of microRNAs, long non-coding RNAs, and other molecules acting as potential disease biomarkers and putative therapeutic targets [135]. Mass spectroscopy is commonly employed to feature EVs due to the large quantity of information provided. The technique is used to determine EV components in terms of lipids, proteins, and metabolites. Liquid chromatography-electrospray ionization tandem mass spectrometry (LC/ESI-MS/MS) is the most common method for EV characterization. EVs are ionized, and their components are separated according to their mass/charge (m/z) values [136,137].

Cancer cell metabolism is significantly altered due to active cell proliferation. In particular, the alterations of cancer metabolism include faster glycolysis and lactic acid production and upregulation of nucleotide synthesis, which consequently lead to exosomal metabolome changes. Ultra-performance liquid chromatography-tandem mass spectrometer (UPLC-MS/MS) was used by Puhka and co-workers to successfully determine over 100 metabolites in the isolated EVs from prostate cancer patients before and after prostatectomy [138].

Paolino et al., driven by the knowledge of the correlation between psoriasis and the increased amounts of high-density lipoproteins and apolipoprotein A1 and an augmented cardiovascular risk, investigated the lipid content of EVs in psoriasis patients using targeted and untargeted liquid chromatography-mass spectrometry (LC-MS) approaches. Interestingly, they suggest that the determination of lipid changes in patient EVs supports the diagnosis and foresees drug response in psoriatic patients [139].

The analysis of EV RNA seems to be particularly interesting as a non-invasive cancer test with high sensitivity and specificity. Multiple RNA types were identified in EVs, including: microRNA (miRNA), ribosomal RNA (rRNA), transfer RNA (tRNA), long non-coding RNA (lncRNA), piwi-interacting RNA (piRNA), small nuclear RNA, and small nucleolar RNA (snoRNA). Messenger RNA (mRNA) has also been identified in EVs as a functional regulator of target cell behavior [140]. Conley and colleagues used next-generation sequencing to demonstrate that EVs carry tumor-specific alterations and can be analyzed as a sample of cancer cell genetic assets [141]. It is important to underline that the EV isolation method influences RNA measurements. Furthermore, as it is currently not possible to sequence RNA in single EVs, the RNA copy number must be considered the average RNA copy number of a pool of EVs [134]. A comparison of the pros and cons of each different isolation method is provided in Table 1 below.

## 2. RBC-Derived Extracellular Vesicles (RBCEVs): Biogenesis and Composition

Human red blood cells (RBCs) are terminally differentiated, enucleated, and very versatile cells that have been used for drug delivery systems since the end of the last century. Most of their applications as drug delivery systems have been extensively reviewed in [142,143,144]. RBCs have a flexible biconcave shape with a diameter of 7.5–8.7 μm and thickness of 1.7–2.2 μm, their membrane mainly contains 60% phospholipids, 30% cholesterol, and 10% glycolipids [145]. Moreover, the RBC membrane also contains various proteins, such as peripheral and integral proteins, which can be classified into three groups: cytoskeletal proteins (e.g., spectrin, actin, protein 4.1), integral structural proteins (e.g., band 3, glycophorins), and anchoring proteins (e.g., ankyrin, protein 4.2) [146]. All these features confer on RBCs a high degree of plasticity and elasticity that in vivo allows them to pass across very small capillaries and exert their transporting function, while in vitro they can be exploited for therapeutic purposes. Hemoglobin is the major cytosolic protein in intact RBCs; the cytoplasmic fraction also contains several antioxidant and metabolic enzymes needed for the RBC’s own metabolism. These proteins can release adenosine triphosphate (ATP) and NO into the intracellular environment [147]. RBCs are the major vesicle-secreting cells in blood circulation. During their 120-day lifespan, RBCs lose ∼20% of their hemoglobin content and membrane integrity during physiological vesiculation [148]. The physiological aging of RBCs accelerates vesicle generation. Indeed, vesiculation is one of the most important mechanisms by which RBCs eliminate harmful substances accumulated throughout their lifespan [149,150]. Under normal physiological conditions, RBC-derived EVs constitute 7.3% of EVs in whole blood, indicating that RBCs are one of the main sources of EVs in peripheral blood [151,152].

RBC membrane vesiculation is a homeostatic process activated in response to impaired signaling machinery; it can be induced by ATP depletion, calcium loading, lysophosphatidic acid exposure, oxidative stress, endotoxins, cytokines, complement, and high shear stress [153]. Red blood cell-derived extracellular vesicles (RBCEVs) can be divided into subpopulations, such as exosomes and microvesicles.

Exosomes can be produced during the reticulocyte or erythroid precursor stage and maintained until the mature RBC stage [4]. Erythropoiesis is a long process that starts with a myeloid precursor and ends with reticulocytes maturing into erythrocytes. This terminal differentiation is accompanied by a cellular remodeling that leads to the disappearance of intracellular organelles, the elimination of membrane and cytoplasmic content, and the acquisition of the typical cellular biconcave form. Blank et al. proposed for the first time that exosome biogenesis and secretion contribute to the net loss of the cell surface membrane via selective vesicular membrane secretion [154]. Exosomes derived from reticulocytes are generated via the typical endosomal pathway of nucleated cells after the plasma membrane has invaginated to form the early endosome. The early endosome subsequently matures into a late endosome that evolves into multivesicular bodies carrying intraluminal vesicles [4].

On the contrary, vesicles may form during the normal aging of circulating erythrocytes due to complement-mediated calcium influx, plasma membrane budding, and subsequent vesicle shedding [155]. The biogenesis and release of microvesicles (MVs), which originate by membrane budding, is an integral part of red blood cell physiology and is linked to their maturation and aging. Indeed, by releasing MVs, the erythrocyte can eliminate damaged components that could also trigger hemostatic and immunological reactions [4]. Primarily, the formation of MVs from red blood cells is triggered by damaged hemoglobin, protein oxidation, degradation caused by senescence, and cytoskeletal binding of ankyrin to band 3. Another mechanism involved in the production of microvesicles is the alteration of phospholipid distribution in the lipid bilayer. Indeed, certain enzymes such as scramblase, calpain, and some proteases can be activated by oxidative stress or an increased influx of calcium ions, thus leading to the inhibition of flippase with the consequent exposure of phosphatidylserine, usually expressed in the inner bilayer layer, or leading to the proteolytic degradation of the cytoskeleton and the consequent aggregation of band 3 [4].

### 2.1. RBCEVs Composition

RBCEVs are generally visualized as round vesicles of 100–200 nm and comprise phospholipids, proteins, cholesterol, lipid rafts, hemoglobin, and acetylcholinesterase [150]. Although RBCEVs are derived from RBCs, their membrane compositions and internal contents are not exactly the same. Indeed, the final composition of RBCEVs is supposed to be dependent on the resealing and stimulating conditions. Indeed, EVs produced from RBCs are reported to be different when produced naturally in vivo, released ex vivo during blood bag storage, or produced in vitro by chemical treatments [156]. According to Yang et al., RBCEVs lack cytoskeletal components such as spectrin and actin, whereas they are relatively enriched in connexins and lipid raft markers [157]. Thangaraju et al. compared about 30 papers reporting the composition of RBCEVs obtained after different procedures of isolation/production and in different physiological/pathological conditions [4]; while Chiangjong et al. made a deep comparison of the components of the parental cells (RBCs) and those of the derived vesicles (RBCEVs) [150]. Taken together, these data suggest that the different conditions or production methods can affect the final composition; however, some similarities can be found and are reported in the next paragraph.

Extracellular vesicles are highly enriched with proteins with different functions, such as tetraspanins (CD9, CD63, CD81, and CD82), which are involved in cell penetration, invasion, and fusion events; heat shock proteins (HSP70, HSP90), which are involved in antigen anchoring and presentation; and MVB formation proteins (Alix, TSG101), which are involved in exosomal release [158]. Additionally, cytoskeletal proteins (e.g., actin), denatured hemoglobin, proteins 4.1, 4.2, enzymes such as carbonic anhydrase, anion transport proteins (Band 3), glycoproteins (e.g., CD235a), membrane-associated proteins such as stomatin and flotillin, and CD47, which inhibits phagocytosis by interacting with macrophage signal regulatory protein alpha [159]. Whereas, in the membrane lipid part, we find phosphatidylserine (PS), which is often found in the outer membrane layer and has a role in biogenesis (see next paragraphs), phosphatidylethanolamine, phosphatidic acid, diacylglycerol, and cholesterol [4,150]. The most striking and interesting discovery is that these vesicles have different types of miRNAs inside them, which can be harnessed and directed towards a specific target to exert their action as modulators of gene expression [4,160]. Among nucleic acids, DNA is obviously lacking, while miRNAs are present in huge amounts [161]. About 78 different miRNAs were found, with miR-125b-5p, miR-4454, and miR-451a being the most abundant [160,161]. On this basis, Sun et al. proposed that under certain conditions, RBCEVs can send miRNA to recipient cells to exert their function [162].

### 2.2. RBCEVs Production under Physiological and Pathological Conditions

RBCEVs are naturally produced during the RBC’s life and are released into circulation, where they can interact with numerous tissues and/or cells to influence their functions. RBCEVs can play roles both in physiological and pathological conditions, but if the RBCEV level in healthy states is assumed to be normal, their level is supposed to be raised with aging or under other diseased or stressed conditions [157].

The first physiological function that has been discovered is during the maturation of RBCs. Indeed, they remove excess proteins and membranes such as transferrin receptors, acetylcholinesterase, and hemoglobin via vesiculation processes during the passage from reticulocyte to mature RBC [163,164]. RBCEVs can protect RBCs by clearing dangerous molecules and preventing their early clearance from circulation [165,166]. In addition, RBCEVs partly inherit the role of RBCs. RBCEVs are critical for communicating with endothelial cells to regulate NO and O2 homeostasis. Indeed, under storage conditions, RBCEVs react faster with NO than intact red blood cells, causing strong vasoconstriction [147,167]. RBCEVs can affect a variety of immune cells [168,169,170]. RBCEVs can promote the production of pro-inflammatory cytokines (interleukin (IL)-2, -7, and -15) and tumor necrosis factor-alpha (TNFα) by interacting with macrophages [170]. RBCEVs also increased proliferation in CD4+ and CD8+ T cells by influencing antigen-presenting cells [168]. Finally, RBCEVs may also have a role in coagulopathy [4]. Some authors have suggested that PS exposed to the external membrane layer can contribute to coagulation by triggering the intrinsic pathway. However, other authors reported an anti-coagulant role of RBCEVs mediated by their interaction with protein S, protein C, and fibrinogen [171,172].

As mentioned before, RBCEVs have been known as key regulators of various physiological and pathological processes, including coagulation, inflammation, and also atherosclerosis and thrombosis [173]. Moreover, there is strong evidence that the plasma concentrations of RBCEVs are elevated in the development of cardiovascular-related disease, which can further lead to vascular dysfunction [174]. Furthermore, RBCs infected by *Plasmodium* falciparum are able to transfer DNA (drug resistance and fluorescent protein genes, etc.) through the RBCEVs, helping the parasite survive times of stress [175]. RBCEVs in disease states often have pro-inflammatory and pro-coagulant effects. Different studies have shown that elevated levels of the circulating pro-coagulant RBCEV lead to increased thrombotic and hypercoagulable states in sudden nocturnal hemoglobinuria (PNH) and hemolytic disease [176,177]. RBCEVs can finally interfere with NO homeostasis by increasing ROS production, which can also lead to endothelial dysfunction [178].

### 2.3. RBCEVs Production and Isolation for Therapeutic Purposes

RBCs are an excellent source for the production of EVs for drug delivery. RBCEVs have several distinct advantages compared with other types of cell-derived and artificially produced EVs [150]. Synthetic nanovesicles created from biomimetic phospholipid bilayers resulted in several improvements, such as increased solubility, prolonged action, reduced toxicity, and lesser adverse effects, by mimicking EV properties [179,180]. However, the issues limiting the utility of those synthetic nanocarriers are immunogenicity and higher clearance by phagocytic cells [181]. Conversely, RBCEVs have been proven extremely safe thanks to their higher biocompatibility compared to synthetic ones. In addition, compared to EVs derived from nucleated cells, RBCEVs have a lower risk of horizontal gene transfer because RBCs lack both nuclear and mitochondrial DNA; thus, RBCEVs are particularly advantageous for delivering genetic material, e.g., RNA molecules and long mRNA molecules [182]. Furthermore, RBCEVs can escape macrophage clearance through the binding of CD47 to inhibitory receptor signal regulatory protein α SIRPα), thus preventing RBCEV clearance via endogenous elimination [159]. Finally, RBCEVs produced by O blood type cells may be used in allogeneic individuals, and the blood cells are easy to obtain from donors. Thus, RBCEVs represent an excellent delivery system for carrying drugs to cellular targets with cost-effectiveness, non-immunogenicity, and high stability and biocompatibility [150].

Nevertheless, the main limitations remain the way of scaling up RBCEVs production to increase the yield and the way to load cargo into EVs. These will be discussed in detail in the following paragraphs.

#### Methods for Scaling up RBCEVs Yield

Several stimuli have been shown to stimulate RBCEVs; Chiangjong and colleagues summarized them very well in a table reporting a list of chemical reagents, oxidative molecules, and storage conditions for RBCEV production [150]. In the next sub-paragraphs, we will discuss the most used methods for producing high amounts of RBCEVs in vitro or ex vivo; these have been outlined in Figure 4.

Among chemical methods, the most used is the in vitro stimulation with calcium ionophore. Usman and colleagues provided a lab-based approach to treat isolated RBCs with calcium ionophores that can stimulate the release of RBCEVs [182]. The protocol also envisaged the purification by means of several rounds of low-speed centrifugations, filtration, and ultracentrifugation steps. This appeared to be a feasible, cost-effective, and high-yielding approach, thus making it one of the most used in laboratory settings. Large-scale amounts (10^13^–10^14^) of RBCEVs can be isolated from RBCs (per unit, 200 mL of blood) when treated with calcium ionophore, which is thus a promising and scalable strategy to obtain EVs in vitro.

Calcium ionophore is supposed to induce vesiculation by activation of PS exposure to the outer surface membrane, thus leading to membrane budding and microvesiculation [150]. Other chemicals can mimic the same process, such as phorbol 12-myristate 13-acetate and lysophosphatidic acid [150].

Another method and/or process that induces vesiculation is the induction of oxidative stress, for example, by means of tert-butyl hydroperoxide, which leads to increased osmotic fragility and Hb oxidation [183].

Finally, it has also been reported that long-term storage, like in blood banking conditions, can stimulate the production of RBCEVs [150,184]. This has obvious consequences in transfusion medicine.

Nowadays, it is not completely understood whether and how different kinds of stimuli can affect RBCEVs’ properties and composition, so characterization is an important step at the end of the production.

In addition to chemical methods, there are different so-called “physical” vesiculation methods that encompass several different methods that can mimic “shear stress”. One of the most used physical vesiculation methods is extrusion, but recently other methods have been proposed.

Gangadaran and co-workers optimized a scaling-up strategy to produce RBCs exosome mimetics (RBC-EMs) based on extrusion [185]. RBC-EMs produced by this method have similar characteristics as RBC exosomes. To obtain RBC-EMs, RBCs were purified from fresh blood and diluted in phosphate-buffered saline. The diluted RBCs were passed through a 1 µm polycarbonate membrane four times by using an extrusion set. After this physical stress, the obtained samples were purified by ultracentrifugation as before. This seems another promising and scalable strategy to obtain RBCEVs and allowed 130-fold higher production yield in terms of particle numbers compared to native exosome release.

Recently, Erytech Pharma patented a novel vesiculation method to produce large-scale RBCEVs also based on “shear stress”. In particular, after the loading procedure (see the next paragraph for a focus on loading procedures), they put loaded RBCs under “strong agitation” for several hours, and this induced the production of RBCEVs. This method will be discussed more deeply in the section dedicated to patents (see “State of the art of the technology from the industrial side”).

More recently, other authors proposed another method to induce mechanical stimuli for vesiculation, which is Piezo1 stimulation [186]. Briefly, Sangha and collaborators found that treating 6% hematocrit RBCs with 10 µM piezo1 agonist yoda1 for 30 min maximized RBCEV yield until 10^12^ particles/mL. This paper was available as a pre-print at the time of its writing.

### 2.4. Methods for Cargo Loading in RBCEVs

Growing evidence shows that RBCEVs can not only deliver biological information but also different kinds of drugs, nucleic acids, and proteins. Nevertheless, another limitation is exactly the way cargo is loaded. Han et al., in 2021, revised the whole methods for loading cargo into different kinds of EVs. He divided the methods between “cell-based” and “non-cell-based” [187]. Cell-based methods, also called pre-loading, are based on the indirect encapsulation of therapeutic cargo into the donor cell before the production of EVs. In this approach, different cargoes can be encapsulated into the donor cells essentially by simple incubation and/or transfection. During vesiculation, cargoes are then packaged into EVs and ultimately delivered to recipient cells for therapeutic use [188]. This method provides a convenient and effective way for loading biological materials and drug therapies into EVs [187]. On the contrary, the non-cell-based loading approach involves the direct loading of chemical or biomolecules into already-produced EVs and can be performed through electroporation, sonication, incubation, and/or transfection [189]. Thus, non-cell-based EV loading methods incorporate therapeutic cargo into EVs after isolation and/or production, and for this reason, it is also known as “post-loading”. Different siRNAs, miRNAs, proteins, CRISPR/Cas9, hydrophobic compounds, and anticancer drugs can be loaded into EVs through non-cell-based loading [187]. These loading methods can be further classified into passive loading and active loading [187]. Passive loading involves loading the therapeutic cargo into EVs through diffusion, whereas active loading consists of the disruption of EV membranes through electroporation or sonication, allowing entry into the EVs.

In this review, which is focused on RBCEVs, we prefer to talk about pre-loading and post-loading methods. As mentioned, RBCs have been used longer for drug delivery [143,144,190]; indeed, they possess numerous features that make them ideal candidates for this purpose: (i) are biodegradable; (ii). are available in large quantities; (iii) can circulate for long periods of time (months); (iv) have a large capacity; (v) are not toxic or immunogenic; (vi) have a long in vivo life span; and (vii) several procedures exist to encapsulate a wide range of molecules inside them. Regarding this last point, many loading methods (electroporation, drug-induced endocytosis, osmotic pulsing, and hypotonic hemolysis) have been set up over the years and are mostly based on the transient opening of pores across the cell membrane, as detailed in Rossi et al. and Magnani et al. [190,191].

RBCs possess, in fact, the remarkable capacity for reversible shape change and for reversible deformation, allowing the opening of pores (20–50 nm in diameter) large enough to be crossed by externally placed macromolecules. Among the above-mentioned methods, the hypotonic hemolysis one is what allows for obtaining engineered RBCs with the most suitable characteristics for biomedical applications. In turn, it includes different procedures, such as dilutional, preswell dilutional, and dialysis ones, which can be opportunely selected by the researchers according to them. However, all these procedures are based on the same physical-chemical features of RBCs. When placed in the presence of a hypotonic solution, RBCs increase in volume, and their morphology is converted to spherocytes; since RBCs have little capacity to resist volume increases when placed in solutions of appropriate mOsm/kg, the membranes rupture with the formation of large pores, permitting the influx of molecules of interest. By raising the salt concentration to its original level, the membranes can be resealed and the added substances entrapped in erythrocytes [192,193]. It must be emphasized that when a procedure moves from the laboratory to the clinic, the availability of appropriate equipment based on an appropriate method becomes very important. To date, two companies, ERYtech Pharma S.A. and EryDel S.p.A., have ongoing phase III clinical trials (ClinicalTrials.gov identifiers: NCT03674242 and NCT03563053, respectively) based on engineered RBCs obtained by dialysis or the preswell dilutional method, respectively. Their equipment allows them to carry out fully automated loading procedures in perfect sterility and apirogenicity conditions, as is needed in clinical use.

Regarding the post-loading methods, the most used for RBCEVs are electroporation and transfection. Both of them have been widely used for the encapsulation of nucleic acids. Iconic examples are represented by Usman et al. that used electroporation [182] and Peng et al. that preferred the transfection method [194]. Several other examples are reported in the next paragraph, which focused on the therapeutic applications of RBCEVs.

The different cargo methods are outlined in Figure 5.

### 2.5. Recent Development in Therapeutic Application of RBCEVs: From the Lab Side to the Industry Side

As discussed, RBCEVs possess several features that make them particularly suitable for therapeutic applications: (i) blood is easily accessible from blood banks, thus RBCs can be produced from blood at large scale and at low cost; (ii) autologous transfusion, from their own donor (or patient), or allogenic transfusion, from a universal 0-donor, of RBCEVs, are possible; (iii) RBCEVs are safer compared with other cell-derived EVs because they lack DNA, minimizing the risk of horizontal gene transfer; (iv) RBCEVs are completely nontoxic and have higher biocompatibility than synthetic EVs; (v) RBCEVs can be frozen and thawed many times without affecting their integrity and efficacy [158]. In this regard, a publication recently investigated different buffers and conditions to allow a longer stability of frozen EVs [195]. The authors demonstrated that EVs, resuspended in suitable buffers, can be stable for up to 2 years. Thus, RBCEVs can be developed into stable pharmaceutical products in the near future (Figure 6). This is probably the main feature that attracted the attention of several biotech and pharma companies.

#### 2.5.1. From the Lab Side

In 2010, Chang and co-workers demonstrated the ability of RBCEVs to efficiently deliver ultra-small superparamagnetic iron oxide particles into human bone marrow mesenchymal stem cells for cellular magnetic resonance imaging in vitro and in vivo [196]. The novel method allowed for higher intracellular labeling efficiency and higher biosafety compared with traditional approaches. RBCEVs were shown for the first time to be biosafe and they can be used as potential delivery vehicles for clinical applications due to their autologous property; this study also gave rise to a patent that is cited in the next section.

But it was in 2018, that the most pioneering study was made by Usman and colleagues, where an efficient delivery system was developed for RNA-based therapeutics using RBCEVs [182]. Small and large RNAs, e.g., antisense oligonucleotides (ASOs), and large RNAs, such as mRNA, were electroporated into RBCEVs and transported to target cells in both solid and liquid tumors. Briefly, microRNA-125b-ASO-loaded RBCEVs significantly reduced both breast tumor growth by intratumoral injection and suppressed acute myeloid leukemia (AML) progression by systemic administration. In addition, genome-editing effects were also observed when RBCEVs were loaded with Cas9 mRNA and guide RNAs. The delivery efficiency was higher, and far less cytotoxicity was observed as compared to other commercial transfection reagents.

Moreover, exosome mimetics (EMs) were produced from red blood cells (RBCs), and the radiolabeling of the RBC-EMs for in vivo imaging was analyzed [185]. Engineered EMs from RBCs were produced on a large scale by a one-step extrusion method and further purified by density gradient centrifugation, the resulting RBC-EMs had a 130-fold greater yield compared to natural nanovesicles generated from RBCs and displayed enhanced in vivo biodistribution. RBC-EMs were labeled with technetium-99m (99mTc). The results demonstrated a simple yet large-scale production method for a novel type of RBC-EMs, which can be effectively labeled with 99mTc, and feasibly monitored in vivo by nuclear imaging. It shows that the RBC-EMs may be used as in vivo drug delivery vehicles [185].

In 2019, RBCEVs were applied in another study in which lipophilic drugs, such as camptothecin, were packaged within RBCEVs and administered to lung carcinoma cells, showing an improvement in targeted delivery when compared with synthetic lipid-based nanocarriers [197].

In 2020, Zhang and colleagues demonstrated that RBC-derived EVs loaded with miR-155 showed an excellent protective effect against acute liver failure, while those loaded with doxorubicin or sorafenib showed significant therapeutic effects against hepatocellular carcinoma without systemic toxicity in mice [198]. In the same year, other authors isolated RBCEVs from subjects infected by *Plasmodium falciparum* and loaded them with the antimalarial drugs atovaquone and tafenoquine [199]. They observed that the free drug was less effective than the RBCEV-loaded one, indicating that RBCEVs can potentially increase the efficacy of several small hydrophobic drugs used for the treatment of malaria.

In 2022, Jayasinghe and co-workers conjugated RBCEVs with several peptides and/or antibodies for targeted delivery of cargoes to cancer cells [200]. They conjugated RBCEVs with a cyclic peptide to specifically target CXCR4 or with a monoclonal antibody anti-CD33 to promote the specific binding and uptake of the conjugated EVs by leukemia cells expressing the corresponding receptors. CXCR4-conjugated RBCEVs were loaded with the pro-apoptotic peptide KLA, demonstrating that these were able to significantly suppress leukemia burden and increase survival in a leukemia xenografted mouse model. Antibody-conjugated RBCEVs were also used to deliver RNA antisense oligonucleotides to knock down FLT3 and miR-125b in cell lines and in patient-derived xenograft models of leukemia [200]. This study demonstrated for the first time that peptide/antibody-conjugated RBCEVs are biocompatible and non-immunogenic and can be used for targeted delivery of therapeutic peptides and RNAs for potential clinical applications. Finally, a novel nanocarrier composed of RBCEVs, surface-linked with doxorubicin using glutaraldehyde, was developed by [7]. The results demonstrated, once again, that drug-loaded RBCEVs could exert superior anticancer activity than free drug, both in vitro and in vivo.

In a very recent study, authors from the same group as Usman et al. [182] re-proposed RNA-loaded RBCEVs for potential immunotherapy [194]. In detail, they loaded the previous 3p-125b-ASO or a novel RIG-I agonist, namely an immunomodulatory RNA. The authors showed that the two agonists stimulated the RIG-I pathway and induced cell death in both mouse and human breast cancer cells. Significant suppression of tumor growth, coupled with increased immune cell infiltration mediated by the activation of the RIG-I cascade, was observed also in vivo after multiple intratumoral injections of RNA-loaded RBCEVs. Finally, they proposed also a targeted delivery using RBCEVs coupled with EGFR-binding nanobody, administrated intrapulmonary to mice, to facilitate the accumulation of RBCEVs in EGFR-positive breast cancer cells.

The main applications reported in the literature have been summarized in Table 2.

#### 2.5.2. To the Industry Side

The first industrial exploitation of RBCEVs-like particles belongs to a researcher at the University of California. In particular, they produced RBC membrane-camouflaged nanoparticles by first producing RBC membrane-derived vesicles by hypotonic treatment and extrusion of RBCs, which were further combined with a polymeric nanoparticle core to produce RBC-derived nanoparticles (Patent Application Publication No. US 2013/0337066A1, Membrane encapsulated nanoparticles and method of use, 2013. Related patent documents: EP2714017 CN103857387 CA2873404 DK2714017 ES2685333 WO/2013/052167 EP3412282 JP2014518200 US20130337066). The inventive nanoparticle comprises (a) an inner core comprising a non-cellular material and (b) an outer surface comprising a cellular membrane derived from RBCs. These nanoparticles were tested in several applications, such as eliciting an immune response, and treating or preventing diseases or conditions, such as neoplasms or cancer, or diseases or conditions associated with cell membrane insertion of toxins. Later, Cellics Therapeutics invented another application of the aforementioned particles: a biomimetic toxin nanosponge that functions as a toxin decoy in vivo. These nanosponges absorb membrane-damaging toxins and can potentially treat a variety of injuries and diseases caused by pore-forming toxins (US 2017/0095510 A1, use of nanoparticles coated with red blood cell membranes to treat hemolytic diseases and disorders, 2017).

In 2019, Le et al. from the City University of Hong Kong proposed for the first time the use of native RBCEVs for gene therapy (US 2019/0054192 Isolation RBCEVs form RBCs for gene therapy, 2019. Related patent documents US20190054192 CN109402176). The invention comprises the purification and electroporation of the RBCEVs and applying the RNA-loaded EVs to target cells. Briefly, they proposed to stimulate the EVs’ production with calcium ionophore, followed by isolation by ultracentrifugation, and finally electroporation to load nucleic acids. Moreover, Minh Le and colleagues from the National University of Singapore and Cornell University developed a method of delivering nucleic acids to immune cells (e.g., T cells) using RBCEVs. These loaded immune cells can be used as immunotherapy to treat cancer (WO 2021/194425 A1, method of delivering nucleic acid to immune cells using RBCEVs, 2021. Related patent documents EP2021777014).

Some years later, researchers at the Imperial College in London proposed the use of targeted delivery of thrombolytic drugs to blood clots using RBCEVs platform WO 2021/123799 A1, red blood cell-derived vesicle, 2021. Related patent documents EP4076481). The invention also extends to a method of: (i) contacting a red blood cell with a hypotonic solution to produce a red blood cell ghost, and (ii) encapsulating an active agent using the red blood cell ghost to thereby produce a red blood cell-derived vesicle comprising an encapsulated active agent. Finally, step (iii) may further comprise extrusion through a filter at least once to produce EVs from ghost RBCs. Using this pre-loading method, RBCEVs have been encapsulated with thrombolytic drugs such as tissue plasminogen activator (tPA), which can further be used to treat ischemic strokes, myocardial infarction, and pulmonary embolism. RBCVs protect thrombolytic agents in the blood circulation, leading to improved stability and prolonged half-life, and temporarily suppress thrombolytic activity, leading to reduced hemorrhagic side effects.

In the same year, Carmine Therapeutics developed next-generation, non-viral based gene therapies to treat a wide range of diseases using RBCEVs (WO 2021/145821 A1, nucleic acid-loaded red blood cells extracellular vesicles 2021. Related patent documents WO/2021/145821 CA3164176 KR1020220127851 CN115151277 EP4090373 NZ789881 JP2022542905). The company proposed a method of isolating calcium ionophore-induced RBCEVs (as before) and further loading these vesicles with nucleic acids (DNA/RNA) by several methods comprising both electroporation and transfection. These platforms showed the ability to carry DNA or RNA payloads ranging from 20 bp to >30 kb and much more. Thus, by using RBCEVs as a drug delivery system, the limitations of viral-based gene therapies, such as immunogenicity, small payload capacity, and manufacturing challenges, can be overcome.

Finally, at the end of the year, Erytech Pharma also filed a patent application for the development of drug delivery RBCEVs from preloaded RBC (WO 2021/228832 A1, red blood cell extracellular vesicles (RCEVS) containing cargoes and methods of use and production thereof, 2021). First, red blood cells were loaded with the desired cargo by “hypotonic encapsulation” and then vesiculated by “strong agitation”. The cargo can be comprised of nucleic acids, proteins, small molecules, or components of a gene editing system, including CRISPR/Cas9. These loaded RBCEVs may be used to treat a variety of diseases and disorders, including autoimmune disorders, cancers, cardiovascular diseases, gastrointestinal diseases, genetic disorders, and inflammatory diseases.

To the best of our knowledge, these are the updates at the time of writing.

## 3. Conclusions and Future Perspectives

Extracellular vesicles, in general, and extracellular vesicles derived from RBCs, are very promising tools in the clinical field. As we mentioned, RBCEVs can act as biological carriers, thus transporting several molecules and cargoes for both therapeutic and diagnostic purposes. The advantages of using EVs over synthetic transport systems are manifold. Indeed, they present greater stability in the bloodstream due to their natural membrane composition, better cargo protection due to their protein–lipid architecture [201], and better biocompatibility, allowing efficient permeability between biological membranes [182]. Especially, RBCEVs present several features that make them particularly suitable for possible clinical applications compared to other types of carriers. RBCEVs are safer when compared to other EVs as RBCs have no nucleus and mitochondria and therefore do not contain genetic material, whereas EVs from other nucleated cell types pose a risk of horizontal gene transfer. Furthermore, RBCEVs are non-toxic and non-immunogenic when transferred to the right blood group, unlike common vectors such as adenoviruses, lentiviruses, nanoparticles, and other synthetic EVs. Finally, blood banks are easily accessible for their production, and the obtained RBCEVs can be frozen and thawed for many cycles without losing their efficacy and integrity [158], thus allowing injectable formulations to be prepared.

A proposed platform for RBCEVs’ exploitation has been proposed in Figure 6. On this platform, the blood can be taken from the same patient (autologous) or from a universal or compatible donor (heterologous). Then, the RBCs are purified and processed for vesiculation and cargo loading under GMP conditions. The obtained loaded RBCEVs can be aliquoted, frozen, and stored for the next injection of the patient(s). It can be hypothesized that up to 100–1000 doses can be obtained from a single blood unit.

Despite all these advantages, there are still important limitations regarding: (i) the lack of a standardized method for their isolation; (ii) the limited efficiency in the loading of the drug of interest; and (iii) the large-scale production at a clinical level. To overcome the isolation/production problems, some authors have proposed in vitro chemical methods to stimulate red blood cells to release a large number of RBCEVs, thus providing a possible strategy to obtain EVs [150,182], while other authors have proposed lab-scale production by physical methods, such as extrusion [185] or vesiculation by agitation (WO2021228832A1, red cell extracellular vesicles (RCEVs) containing cargoes and methods of use and production thereof). Regarding vesicle loading, two different approaches have been basically proposed: the molecule of interest can be loaded into the cells before the biogenesis of EVs (pre-loading method) or after their formation by electroporation (post-loading method).

Once these processes are optimized, the applications will be manifold. From gene therapies to RNA-based therapies, from enzyme replacement therapies to the new trends in genome editing. Despite this, there are still many unanswered questions about how they can work in vivo, such as: How are EVs internalized in recipient cells? How is the cargo released in them? How does the vesicular cargo affect the physiology of the recipient cell? What may be the most advantageous routes of administration, and which organs might be possible targets? Finding answers to all these questions will pave the way for the long-awaited transition from the lab to the clinic.

## Figures and Tables

**Figure 1 pharmaceutics-15-00365-f001:**
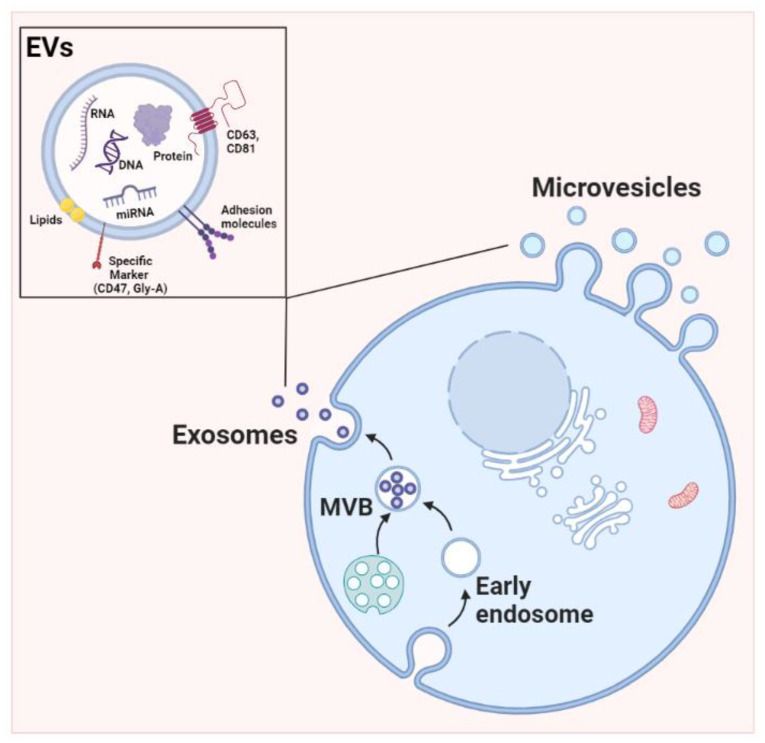
Scheme of biogenesis and typologies of cargoes of extracellular vesicles. Extracellular vesicles (EVs) generated via different mechanisms. Exosomes are produced from inward budding of the endosomal membrane. Microvesicles generated by outward budding of the plasma membrane. The outer EV membrane contains lipids and transmembrane proteins. EVs are also enriched in microRNAs, mRNAs, and proteins that mediate systemic effects. EVs are also known to contain metabolites and mitochondrial DNA. Created in Biorender.com, accessed on 6 December 2022.

**Figure 2 pharmaceutics-15-00365-f002:**
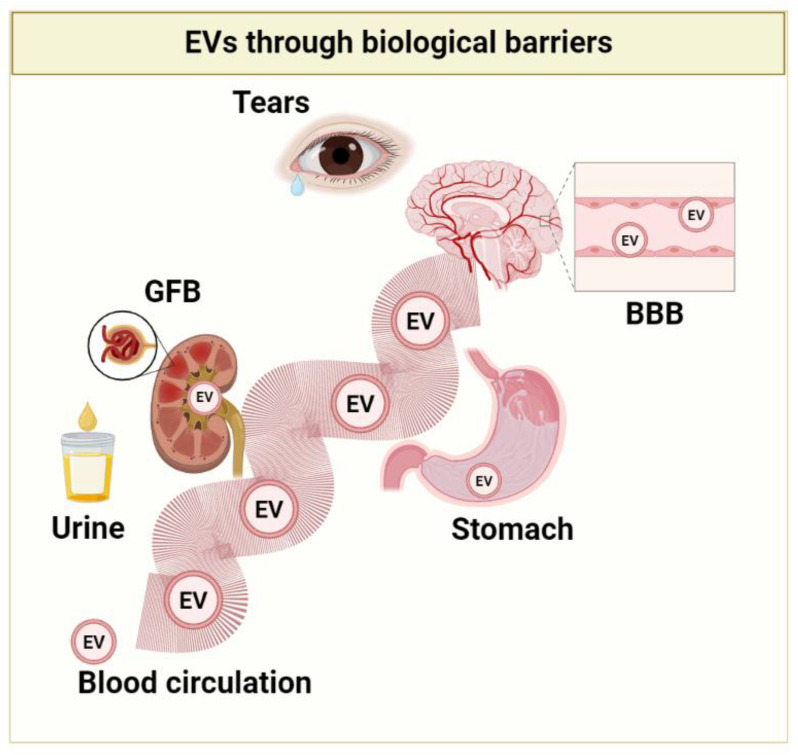
Summary of the different biologic barriers penetrated by EVs. GFB: glomerular filtration barrier; BBB: blood–brain barrier. Created in Biorender.com, accessed on 6 December 2022.

**Figure 3 pharmaceutics-15-00365-f003:**
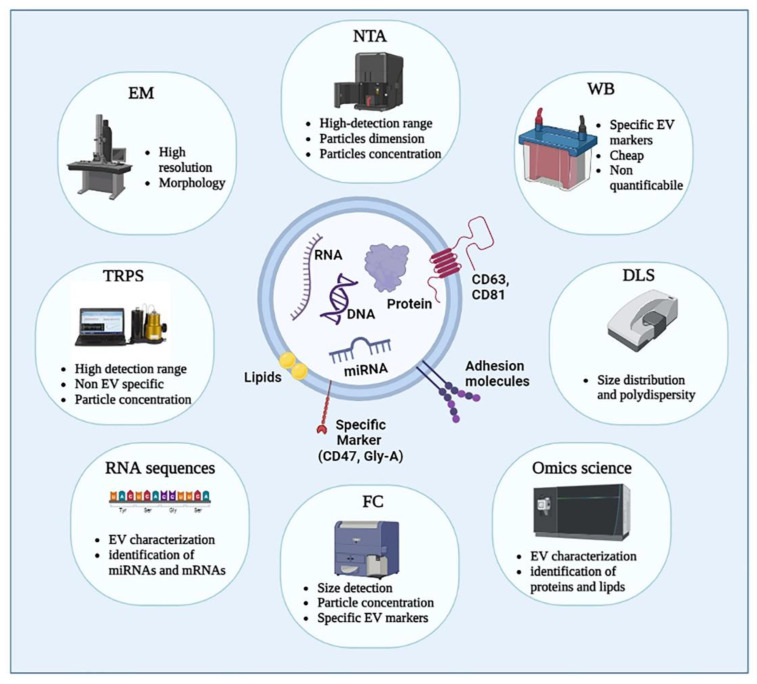
Summary of the different methods for characterization and quantification of EVs. The characterization and quantification of EVs are performed through different techniques such as nanoparticle tracking analysis (NTA), flow cytometry (FC), Western blot (WB), transmission electron microscopy (TEM), and scanning electron microscopy (SEM). TEM analyzes EV diameters and morphology. Dynamic light scattering (DLS) determines EV size distribution and polydispersity by detecting the diffusion coefficient of the scattering EVs. NTA detects EV dimensions and concentrations through scattered light measurement and Brownian motion. RNA sequences and mass spectrometry permit cargo characterization. In particular, FC can be employed for size detection, particle concentration, and EV-specific markers. Tunable resistive pulse sensing (TRPS) only measures particle concentration. Created in Biorender.com, accessed on 6 December 2022.

**Figure 4 pharmaceutics-15-00365-f004:**
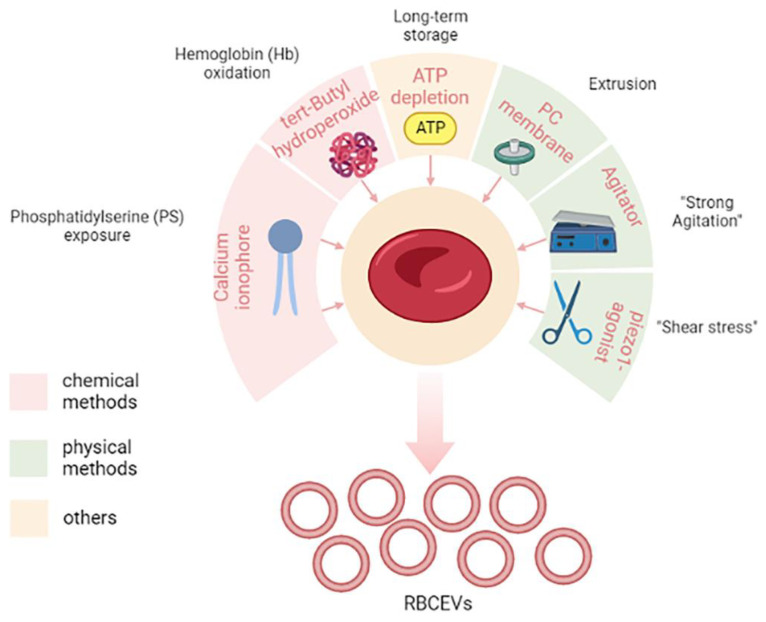
Methods for increasing the yield of RBCEVs production. In the figure, different chemical and physical methods are shown. In particular, molecules and/or mechanical stimuli are depicted together with their effect on the cells that give rise to RBCEVs release. Long-term storage is a particular condition that cannot be included in either the chemical or physical methods. Created in Biorender.com, accessed on 5 December 2022.

**Figure 5 pharmaceutics-15-00365-f005:**
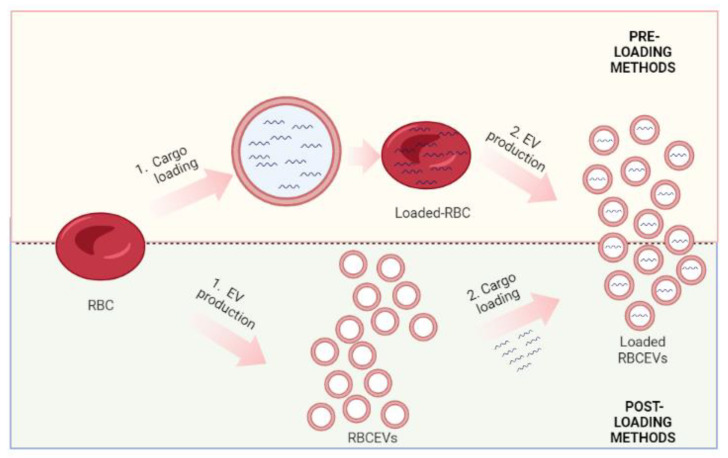
Methods for cargo loading in RBCEVs. Loading methods have been divided into two main classes, namely pre-loading and post-loading methods. In the first, the cargo is loaded into the donor cell before the vesiculation process. While, in the second method, the cargo is loaded into the already produced vesicles. Created in Biorender.com, accessed on 9 January 2023.

**Figure 6 pharmaceutics-15-00365-f006:**
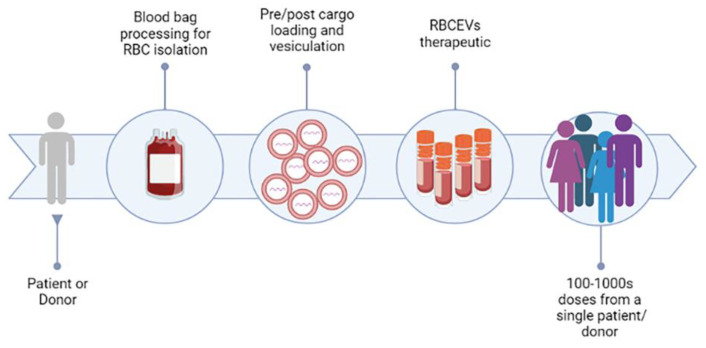
RBCEVs platform for clinical exploitation. The figure represents the road for the clinical exploitation of RBC-derived EVs, which could be achieved in the near future, and shows some of the advantages of this promising strategy. Created in Biorender.com, accessed on 14 December 2022.

**Table 1 pharmaceutics-15-00365-t001:** Pros and cons overview of principal EV isolation methods.

EV Isolation Methods		
Method	Pros	Cons
Differential ultracentrifugation (dUC)	Cost-effective [1] Well established and widely employed [134]	Lipoprotein contamination [134] Time consuming [1] Massive EV aggregation and damaging [1]
Density gradient UC	EV subpopulations distinction [1]	Need for operator high training [1] Low final purity [1] Small sample volumes can be processed [1]
Size exclusion chromatography (SEC)	No high-density lipoprotein contamination [111] Serum albumin removal [111] Fast and cheap method [134] EV integrity preservation [1]	Difficulties in EV subpopulations distinction [111] Co-isolation of non-EV particles above the cut-off [1,134]
Immuno-capture	Selective EV isolation from different cell types [111] Easy method, simple instrumentations [111]	Expensive ligands [111] Need for optimization of a mild elution process [111]
Ultrafiltration (UF)	Simple and cheap method [1,111]	Protein contamination [111] Vesicle damaging [1]
Asymmetrial flow field-flow fractionation (AF4)	Gentle fractionation [111] Provides a large range of size-based separation [106] No lipoprotein contamination [111]	Small sample volumes can be processed [106] Co-isolation of particles with the same hydrodynamic size [106]

**Table 2 pharmaceutics-15-00365-t002:** Main findings on RBCs-based EVs as a drug delivery carrier.

Reference	EV Production Method	Cargo-Loading Method	Cargo	Application	In Vitro	Pre-Clinical	Clinical
[196]	Chemical method, calcium chloride	Incubation under hypo-osmotic conditions	Ultrasmall superparamagnetic iron oxide (USPIO) particles	Magnetic resonance imaging	X	X	
[182]	Chemical method, calcium ionophore	Post-loading method, electroporation	Antisense oligonucleotides, Cas9 mRNA, and guide RNAs	Cancer therapy	X	X	
[185]	Physical method, extrusion	Post-loading method, incubation	Technetium-99m	In vivo imaging		X	
[197]	Physical method, extrusion	Pre-loading method, hemolysis and incubation	Camptothecin and amphiphilic fluorophore	Cancer therapy	X	X	
[7]	Chemical method, calcium ionophore	Post-loading method, incubation	Doxorubicin	Cancer therapy	X	X	
[194]	Chemical method, calcium ionophore	Post-loading method, transfection and electroporation	Rig-I agonists, small RNAs	Cancer therapy	X	X	
[198]	Chemical method, calcium ionophore	Post-loading method, transfection	Antisense oligonucleotides, doxorubicin and sorafenib	Acute liver failure, cancer therapy	X	X	
[199]	Isolation of naturally produced RBCEVs	Post-loading method, incubation	Antimalarial drug, atovaquone and tafenoquine	Anti-malarial treatment	X		
[200]	Chemical method, calcium ionophore	Post-loading method, transfection	Peptide, Antisense oligonucleotides, siRNA	Cancer therapy	X	X	

## Data Availability

No new data were created or analyzed in this study. Data sharing is not applicable to this article.

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
