# Peer review of "Extracellular Vesicles as New Players in Drug Delivery: A Focus on Red Blood Cells-Derived EVs"

_pharmaceutics, 2023, doi:10.3390/pharmaceutics15020365_

Round 1

Reviewer 1 Report

Biagiotti et al presented an overview of basic EV characteristics and isolation methodologies, focusing on the application of red blood cells-derived EVs for drug delivery. The manuscript is clear and interesting, and the figures are well-designed. However, some points can be improved.

Major considerations:

1) The abstract could be restructured to contain a brief introduction, the objectives of the review (as described), and a conclusion (following the model suggested by the Journal). 

The same applies to text. I believe it would be very interesting to include a small and brief introduction in the manuscript, before the current section 1, describing the “problem” and the objectives/ parts of the review.

2) Sentence in lines 72-74, about transferring content from MVs. It seems that only MVs contain DNA, mRNA, miRNA, and others that could be transferred; in the description of exosomes, this is not mentioned and is not clear to the reader. Please review.

3) In the last sentence of topic 1.2, the authors indicate that “…specific details will be added to better understand the mechanisms that mediate the vesicle transfer and uptake and their application as tools-mediated drug delivery.”. However, the other topic is more about the potential to be a tool for drug delivery rather than the mechanisms of cell-EV interaction. Please, review.

4) The authors state that “There are many reviews that deal with methods of EV analysis…” (lines 167-168). Please, cite some references from reviews on characterization methods.

5) The content of EVs can change depending on many aspects. Then the characterization of EVs is critical. Despite the classical methods described in the manuscript, to better characterize the EVs, high-throughput strategies for the identification of miRNAs and/or mRNAs (RNA sequencing), proteins (mass spectrometry/ proteomics), lipids (lipidomic), metabolites (metabolomic) and others are also performed. So, I think it would be interesting to include these methodologies in figure 3 or cite them.

6) In the second section of the manuscript I noticed some sentences that are very similar (or identical) to another article (reference 101, Chiangjong et al, 2021). Some of the sentences are on lines 279-282, 286-288, and 394-398. Another sentence is from lines 552-554 similar to reference 107 (Zhang et al, 2018). Please revise the entire manuscript. The sentences should be modified, and/ or the references must be included in all of them.

7) It would be interesting to summarize in a table the advantages and disadvantages of the EV isolation methodologies (Topic: 1.4 Different Extracellular Vesicle Preparations/ Isolations: an overview).

8) In the Topic “2.1 RBCEVs composition”, lines 327- 331, the authors indicate 2 reviews that compared the composition of RBCEVs, Thangaraju et al and Chiangjong et al. But what is the main conclusion about the composition? It would be interesting present a brief idea of what the articles showed.

9) Figure 5 could be cited over the text, for example, on lines 492-493. In addition, this figure is not so clear. In the pre-loading method, it looks like the cargo loading is leaving the RBC (as the EVs), but that is not the case. Also, it is unclear why there is a DNA at the top of the figure.

10) Figure 6 could be cited elsewhere and explored further. It is like an ideal platform for the production of RBCEVs. So, it could be more described in the conclusions and perspectives, for example.

11) EVs can cross biological barriers and reach various organs. In therapy, how to direct the EVs to a certain target organ? Do all EVs have the same potential to go to all organs? This would be something interesting to address, citing examples or as a difficulty to be overcome, as it would be fundamental for therapy or the use of EVs as a drug delivery device.

12) Lines 502 – 516 have no reference. Please, include.

Minor considerations:

- I suggest that the Keywords are written not abbreviated. 

- Line 58: What is ESCRT? Please define the abbreviation in the text.

- Line 168: Is it “paragraph” or topic/ section?

- In the legend of Figure 3, line 175, the authors include “RBC-EV”, but there it's still talking about EVs in general.

- Line 220-221: What is the reference?

- Line 435: What is PS? Please define the abbreviation in the text.

- Lines 432 and 469: 1013, 1014, and 1012 are, respectively, 1e13, 1e14, and 1e12?

- Lines 469-470: The authors state that “This paper was not published at the time of paper writing”. Is it necessary to maintain? It’s a bit confusing. Maybe include in the other sentence that it is a preprint.

- Line 537: “…,thus RBCs can be produced…”. Is it RBCEVs?

- Line 582: “…RBCEVs infected with Plasmodium falciparum…” or “…RBCEVs derived from RBCs infected with…”?

- Problems with reference format: lines 251-2, 480-1. Please, check.

Author Response

1) The abstract could be restructured to contain a brief introduction, the objectives of the review (as described), and a conclusion (following the model suggested by the Journal). The same applies to text. I believe it would be very interesting to include a small and brief introduction in the manuscript, before the current section 1, describing the “problem” and the objectives/ parts of the review.

Response: Thank you for your suggestion. We modified the abstract according to your precious suggestions.

2) Sentence in lines 72-74, about transferring content from MVs. It seems that only MVs contain DNA, mRNA, miRNA, and others that could be transferred; in the description of exosomes, this is not mentioned and is not clear to the reader. Please review.

Response: We reformulated the sentence, including exosomes in the description of possible cargoes.

3) In the last sentence of topic 1.2, the authors indicate that “…specific details will be added to better understand the mechanisms that mediate the vesicle transfer and uptake and their application as tools-mediated drug delivery.”. However, the other topic is more about the potential to be a tool for drug delivery rather than the mechanisms of cell-EV interaction. Please, review.

Response: Thank you to highlight this issue. We have now eliminated the sentence “the mechanisms that mediate the vesicle transfer and uptake”, in accordance with the following described topic.

4) The authors state that “There are many reviews that deal with methods of EV analysis…” (lines 167-168). Please, cite some references from reviews on characterization methods.

Response: We added some references after the indicated statement. For your early convenience we report here;

  • Bioengineering 2019, 6(1), 7;
  • Int J Nanomedicine 2020 Sep 22;15:6917-6934.
  • Pharmacol.,10 November 2021 Sec. Experimental Pharmacology and Drug Discovery
  • Am J Reprod Immunol. 2021 Feb;85(2):e13367.

5) The content of EVs can change depending on many aspects. Then the characterization of EVs is critical. Despite the classical methods described in the manuscript, to better characterize the EVs, high-throughput strategies for the identification of miRNAs and/or mRNAs (RNA sequencing), proteins (mass spectrometry/ proteomics), lipids (lipidomic), metabolites (metabolomic) and others are also performed. So, I think it would be interesting to include these methodologies in figure 3 or cite them.

Response: Thank you for the suggestion. We added a short introduction on the most commonly used strategies to analyse EV content. We also added in the figure 3.

6) In the second section of the manuscript I noticed some sentences that are very similar (or identical) to another article (reference 101, Chiangjong et al, 2021). Some of the sentences are on lines 279-282, 286-288, and 394-398. Another sentence is from lines 552-554 similar to reference 107 (Zhang et al, 2018). Please revise the entire manuscript. The sentences should be modified, and/ or the references must be included in all of them.

Response: We thanks the reviewer for this suggestion, sometimes the similar expression is inevitable, but we tried to rephrase those sentences, like others in the text.

7) It would be interesting to summarize in a table the advantages and disadvantages of the EV isolation methodologies (Topic: 1.4 Different Extracellular Vesicle Preparations/ Isolations: an overview).

Response: Thank you for your precious suggestion! We added a table gathering the principal EV isolation strategies together with their pros and cons.

8) In the Topic “2.1 RBCEVs composition”, lines 327- 331, the authors indicate 2 reviews that compared the composition of RBCEVs, Thangaraju et al and Chiangjong et al. But what is the main conclusion about the composition? It would be interesting present a brief idea of what the articles showed.

Response: We thank the reviewer to make us notice this gap. A new sentence reporting our idea has been added at line 571-573.

9) Figure 5 could be cited over the text, for example, on lines 492-493. In addition, this figure is not so clear. In the pre-loading method, it looks like the cargo loading is leaving the RBC (as the EVs), but that is not the case. Also, it is unclear why there is a DNA at the top of the figure.

Response: Indeed, figure 5 was already cited into the text (please see line 786). Conversely, the reviewer is right when saying that the figure is not so clear. Unfortunately, a wrong version of the figure has been uploaded; a new one has been embedded in the revised manuscript.

 10) Figure 6 could be cited elsewhere and explored further. It is like an ideal platform for the production of RBCEVs. So, it could be more described in the conclusions and perspectives, for example.

Response: Done. The figure has been cited at line 805 and a new sentence(s) has been added in the last part of the review (line 972-977)

11) EVs can cross biological barriers and reach various organs. In therapy, how to direct the EVs to a certain target organ? Do all EVs have the same potential to go to all organs? This would be something interesting to address, citing examples or as a difficulty to be overcome, as it would be fundamental for therapy or the use of EVs as a drug delivery device.

Response: Thank you for your comment. We expanded the paragraph named “EVs benefits: their journey to the different body districts” to answer to your questions. We thank the reviewer for this comment. Some possible applications and target organs are reported in the paragraph 2.5 together with some strategies for a specific targeting to organs or tissues. In addition, a new table has been produced to outline and/or summarize possible strategies to set up therapeutic platforms, applications and/or targeting.

12) Lines 502 – 516 have no reference. Please, include.

Response: References have been added as requested.

RESPONSES TO REVIEWER 1 MINOR CONSIDERATIONS

- I suggest that the Keywords are written not abbreviated.

Response: DONE

- Line 58: What is ESCRT? Please define the abbreviation in the text.

Response:  Endosomal Sorting Complex Required for Transport (ESCRT), we added in the main text

- Line 168: Is it “paragraph” or topic/ section?

Response: It is a topic, thank you for the suggestion. We modified it!

- In the legend of Figure 3, line 175, the authors include “RBC-EV”, but there it's still talking about EVs in general.

Response: We modified, in according with your suggestions

- Line 220-221: What is the reference?

Response: We added the reference in the text. For your early convenience, Pharmaceutics 2019, 11(11), 557.

- Line 435: What is PS? Please define the abbreviation in the text.

Response: It stands for phosphatidylserine, we fixed it.

- Lines 432 and 469: 1013, 1014, and 1012 are, respectively, 1e13, 1e14, and 1e12?

Response: yes, thanks for noting this digit error that has been now corrected.

- Lines 469-470: The authors state that “This paper was not published at the time of paper writing”. Is it necessary to maintain? It’s a bit confusing. Maybe include in the other sentence that it is a preprint.

Response: changed.

- Line 537: “…,thus RBCs can be produced…”. Is it RBCEVs?

Response: In this case, we mean RBC production; a specification has been added.

- Line 582: “…RBCEVs infected with Plasmodium falciparum…” or “…RBCEVs derived from RBCs infected with…”?

Response: changed; we meant RBCEVs from subjects infected with….

- Problems with reference format: lines 251-2, 480-1. Please, check

Response: We fixed the issue, thank you.

Reviewer 2 Report

Overall, this review manuscript provides a clear overview of the focus of the article, which is to examine the nature, features, and applications of extracellular vesicles (EVs) with a specific focus on those derived from red blood cells. It also mentions the inclusion of a summary of considerations for future clinical translation of EVs and their potential industrial exploitation.

To improve the manuscript, some potential suggestions include:

1.     Clarifying the scope of the review in the first sentence. Currently, it mentions "extracellular vesicles" in general, but then focuses on a specific kind of extracellular vesicles, namely those derived from red blood cells. Clarifying this focus from the outset may help the reader understand the context of the review.

2.     Include more specific details on the methods and strategies used to isolate, prepare, and characterize EVs in the article. This can help readers better understand the approaches and techniques used in the review.

3.     Expand on the discussion of the potential industrial exploitation of EVs, as this is mentioned in the abstract but not elaborated upon. Providing more information on this topic could add further value to the review.

4.     Provide a brief overview or summary of the main findings or conclusions of the review in the abstract. This can help readers quickly understand the key takeaways from the article.

5.     Consider adding a sentence or two outlining the main limitations or gaps in the current knowledge that are addressed in the review. This will help the reader understand the relevance of the review in the context of the current state of the field.

6.     A table should be added summarizing the main findings on RBCs-based EVs as a drug delivery carrier.

7.     Consider revising the language to make it more concise and straightforward. For example, the phrase "quick outline of advances" could be revised to simply "overview of advances."

8.     Adding a few keywords or phrases that will help the reader understand the main themes of the review and make it more discoverable through search engines. For example, adding terms such as "nanomedical drug delivery" or "therapeutic applications" may be helpful.

Author Response

  1. Clarifying the scope of the review in the first sentence. Currently, it mentions "extracellular vesicles" in general, but then focuses on a specific kind of extracellular vesicles, namely those derived from red blood cells. Clarifying this focus from the outset may help the reader understand the context of the review.

Response: Thank you for your suggestion. The first 6 lines of the INTRODUCTION have been added with the aim to highlight the main focus of the Review.

  1. Include more specific details on the methods and strategies used to isolate, prepare, and characterize EVs in the article. This can help readers better understand the approaches and techniques used in the review.

Response: We added several technical details to help the scientific community to better frame different approaches. In addition, we provided a short overview on the most common methods employed to characterize EVs.

  1. Expand on the discussion of the potential industrial exploitation of EVs, as this is mentioned in the abstract but not elaborated upon. Providing more information on this topic could add further value to the review.

Response: We thanks the reviewer for this comment; however, we do not totally agree with him. Potential clinical and /or industrial application has been reported in paragraph 2.5 and in part in paragraphs 2.3 and 2.4. Moreover, several sentences have been reported in the conclusions and perspectives section. It’s noteworthy that at the moment none of the applications reached the clinic, but several are the patents deposited to cover this field. Thus, the road for the real industrial exploitation is near to come (as outlined in figure 6). Finally, a new table has been added to recapitulate the applications proposed up to now at in vitro and preclinical levels.

  1. Provide a brief overview or summary of the main findings or conclusions of the review in the abstract. This can help readers quickly understand the key takeaways from the article.

Response: We rewrite the abstract, highlighting the main points of the first, general part, and precisely mentioning the main findings of the second part.

  1. Consider adding a sentence or two outlining the main limitations or gaps in the current knowledge that are addressed in the review. This will help the reader understand the relevance of the review in the context of the current state of the field.

Response: In the first 6 lines of the INTRODUCTION, we added the following sentence: “Specifically, we will frame red blood cells as sources of EVs (RBCEVs) that can address the major requirements for efficient drug-delivery, providing a useful and insightful description of procedures (also patented) to produce RBCEVs, with their advantages and limitations.” In our opinion, this will help the reader understand the relevance of the review in the context of the current state of the field.

  1. A table should be added summarizing the main findings on RBCs-based EVs as a drug delivery carrier.

Response: We wish to thank the review for this nice suggestion. As mentioned before, a table has now been added to the revised manuscript.

  1. Consider revising the language to make it more concise and straightforward. For example, the phrase "quick outline of advances" could be revised to simply "overview of advances."

Response: Thank you! We made the most of your suggestion.

  1. Adding a few keywords or phrases that will help the reader understand the main themes of the review and make it more discoverable through search engines. For example, adding terms such as "nanomedical drug delivery" or "therapeutic applications" may be helpful.

Response: Thank you! We added a few of them in the text.

Reviewer 3 Report

The authors reviewed red blood cells-derived extracellular vesicles (EVs) as a drug delivery in biological application, especially in the manufacture and physical assessment. This manuscript was well prepared, I recommend the acceptance after some revision.

1. The comparsion between EVs and translational nanoparticles could be discussed in a paragraph.

2. Others biological-aspired nanoparticles, such as cell membranes nanoparticles could be discussed with EVs.

3. Some related references could be refer to author's revision ( Advanced Materials, 2022, 34 (25): 2107674. Science Bulletin, 2022, 67 (16): 1611-1613.)

Author Response

  1. The comparsion between EVs and translational nanoparticles could be discussed in a paragraph.

Response: Thank you for your suggestion. We added a brief comparison in the paragraph named “Other actors in next generation drug delivery platforms: taking a glance”

  1. Others biological-aspired nanoparticles, such as cell membranes nanoparticles could be discussed with EVs.

Response: We discussed about them in the paragraph “Other actors in next generation drug delivery platforms: taking a glance”

  1. Some related references could be refer to author's revision (Advanced Materials, 2022, 34 (25): 2107674. Science Bulletin, 2022, 67 (16): 1611-1613.)

Response: Thank you for your suggestion. We added these related references in the new paragraph added.

Round 2

Reviewer 1 Report

Dear authors,

Thank you so much for the answers. The manuscript showed great improvement. Congratulations.